# Gut microbiome composition and strain-sharing in multiplex autism spectrum disorder families

Wenqi Lu[1,2,8], Oscar W. H. Wong [3,8], Jie Zhu [1,4], Sizhe Chen [1,2], Hein M. Tun [1,4,5], Yating Wan[1,2], Zhilu Xu [1,2], Chun Pan Cheung[1,2], Jessica Y. L. Ching [1,2], Pui Kuan Cheong[1], Sandra Chan[3], Samuel Wong[5], Dorothy Chan[6], Francis Ka Leung Chan[1,2,4,7], Qi Su [1,2] ✉ & Siew Chien Ng [1,2,4,7] ✉

Autism spectrum disorder (ASD) is associated with alteration of gut microbiome, but the influence of familial structure on it remains poorly understood. We investigate gut microbiota across 429 children from multiplex families with multiple affected children, simplex families with one affected child, and single-child ASD families, alongside typically developing controls. We found that children from multiplex families exhibit the most distinct microbiome compositions. Cohabiting siblings in ASD families display higher microbiome similarity than those in healthy families, with a clear gradient in strain-sharing rates that is highest in multiplex, intermediate in simplex, and lowest in healthy siblings. This increased sharing involves specific taxa with reported opportunistic pathogenic potential, such as *Eubacterium rectale*, alongside reduced sharing of the commensal bacterium *Bacteroides xylanisolvens*. This suggests that their gut microbiome configurations, which are potentially influenced by shared environmental and host factors, are associated with increased persistence or detectability of specific bacterial strains. Our results underscore the significant contribution of family type to microbial heterogeneity in ASD and provide a hypothesis-generating context for future studies to explore the role of the shared microbial environment in a familial context.

Autism spectrum disorder (ASD) is a group of neurodevelopmental conditions defined as impaired social communication and interactions and restricted repetitive behavior[1–4]. Although the etiology of ASD remains unclear, it appears to involve a complicated interaction of genetic and environmental factors[5,6]. In the past years, different family types of ASD, including multiplex families with multiple affected children and simplex families with only one affected child, have captured the interest of researchers[7–12]. Previous studies indicate that children with ASD have altered gut microbiome composition and function compared with typically developing (TD) children[13,14], and the gut microbiome has been shown to play a crucial role in the gut-brain axis, which may contribute to the neurobehavioral and intestinal

[1]Microbiota I-Center (MagIC), Hong Kong SAR, China. [2]Department of Medicine and Therapeutics, The Chinese University of Hong Kong, Hong Kong SAR, China. [3]Department of Psychiatry, The Chinese University of Hong Kong, Hong Kong SAR, China. [4]Li Ka Shing Institute of Health Sciences, State Key Laboratory of Digestive Disease, Institute of Digestive Disease, The Chinese University of Hong Kong, Hong Kong SAR, China. [5]The Jockey Club School of Public Health and Primary Care, The Chinese University of Hong Kong, Hong Kong SAR, China. [6]Department of Paediatrics, Prince of Wales Hospital, Hospital Authority, Hong Kong SAR, China. [7]Center for Gut Microbiota Research, Faculty of Medicine, The Chinese University of Hong Kong, Hong Kong SAR, China. [8]These authors contributed equally: Wenqi Lu, Oscar W. H. Wong. ✉e-mail: qisu@cuhk.edu.hk; siewchienng@cuhk.edu.hk

dysfunction in ASD[15,16]. Concomitantly, various factors, such as lifestyle and diet, have been widely recognized to influence the human gut microbiome[17,18]. However, the majority of microorganisms are acquired from other individuals[19,20], and understanding the acquisition and development of the gut microbiome is a crucial component for the clinical translation and intervention of microbiome research. So far, studies focused on the transmission of the microbiome highlight the contribution of intra-household transmission by the assessment of the bacteria strain-sharing across individuals, and cohabitation appears to be more strongly linked to the transmission and co-acquisition than age and genetics did[20–22]. These findings sparked our curiosity about whether the gut microbiome in children with ASD would be affected by their cohabitation with siblings among different family types. In this study, we performed metagenomic analysis of fecal samples of 429 children, characterized the gut microbiome composition, and quantified the patterns of microbiome strain sharing in different ASD families.

## Results

### Study characteristics
A total of 429 well-characterized children were recruited in this study. Within this cohort, 312 children represented three types of multi-child families: "multiplex" families (two or more children diagnosed with ASD), "simplex" families (one affected child with at least one unaffected male sibling), and "TD" families (two or more typically developing children and no children with ASD). Another 117 children from only-child ASD families were included for broader comparison. Amongst the 429 children (mean age: 7.05) studied, there were 37 multiplex ASD families (ASD children, $n = 75$, males 74.6%, namely ASD-M), 50 simplex ASD families (ASD children, $n = 50$; males 94%, namely ASD-S; healthy siblings, $n = 54$, Male: 96.3%), 66 TD families (typically developing children, $n = 133$, males 59.6%; namely TD), and 117 only-child families (ASD, $n = 117$, males 87.1%, namely ASD-O). Age, Gender, and the presence of attention-deficit/hyperactivity disorder (ADHD) are significantly different among these groups; thus, these factors were adjusted in the following association analyses. We performed metagenomic sequencing on 429 fecal samples. The bacterial dataset was filtered by 5% prevalence and contained 509 taxa at the species level. For the strain-level profiling, we have detected 368 strains for the strain-sharing assessment.

### Alterations of taxonomic profiling in ASD-M
Bray-Curtis dissimilarity-based principal coordinate analysis (PCoA) revealed that the gut microbiome difference between ASD-M and TD children is larger than that between ASD-S and TD children (PERMANOVA test: $p = 0.001$ and $p = 0.242$, Fig. 1a). When compared to TD children, bacteria alpha diversity and richness were significantly increased in ASD-M ($p = 0.017$ and $p = 0.022$) but not in ASD-S ($p = 0.97$ and $p = 0.64$, Fig. 1b). To further confirm this finding, we further tested the gut microbiome difference between ASD children from ASD-O and TD children, and we found it lies between the ASD-M and ASD-S. This finding indicates the microbiome in ASD-S was closer to TD, but ASD-M was far from TD when compared with ASD-O.

We further analyzed the above findings at the species level. To account for potential confounding factors, all species-level analyses were conducted using MaAslin2 with adjustment for age, gender, and ADHD comorbidity as fixed effects in the statistical models. 18 bacterial species were identified to be associated with ASD-M ($p < 0.05$, FDR < 0.2, MaAslin2, Fig. 1c and Supplementary Data 4), whereas only 3 species were significantly associated with ASD-S ($p < 0.05$, FDR < 0.2, MaAslin2, Fig. 1c and Supplementary Data 9), when compared to TD children, respectively. In detail, a total of 18 bacterial species associated with potential pathogenesis, such as *Coprobacillus cateniformis*, *Alistipes finegoldii*, *Murimonas intestini*, *Romboutsia timonensis*, and *Eisenbergiella tayi*, were enriched in ASD-M. In contrast, 3 beneficial

species *Faecalibacterium prausnitzii*, *Bacteroides xylanisolvens*, and *Agathobaculum butyriciproducens* were enriched in the TD group. For better comparison, we also analyzed ASD children from ASD-O and TD children, but we did not observe similar results like ASD-M versus TD, and no significant differences in microbial taxa were found ($p < 0.05$, FDR < 0.2, MaAslin2, Fig. 1c). Furthermore, although the overall dietary patterns showed no significant differences between groups based on PCA analysis of daily diet intake (Supplementary Fig. 1), we have conducted sensitivity analyses adjusting for dietary factors. First, we performed Kruskal–Wallis tests for overall differences of dietary factors (Supplementary Data 2), followed by pairwise Wilcoxon tests with FDR correction for each significantly different factor (Supplementary Data 3). After incorporating these significant dietary factors into Maaslin2 models, the main findings remained robust. Specifically, 10 and 3 differentially abundant bacterial species were associated with ASD-M and ASD-S, respectively (Supplementary Fig. 2b, h and Supplementary Data 5 and 10), and 2 associated species emerged in the ASD-O group (Supplementary Fig. 2n and Supplementary Data 13). We also conducted a stratified analysis regarding the comorbidity of ADHD by categorizing ASD children into four subgroups: ASD-M (ADHD+), ASD-M (ADHD−), ASD-S (ADHD+), and ASD-S (ADHD−) ($n = 37, 38, 11$, and $39$, respectively). The key finding that ASD-M children had the most distinct gut microbiome from TD children remained significant even when including only participants without an ADHD diagnosis. Notably, ASD-M (ADHD−) children consistently exhibited a higher number of differential bacterial species than ASD-S (ADHD−) children, both before (Supplementary Fig. 2c, i and Supplementary Data 6 and 11) and after adjusting for dietary factors (Supplementary Fig. 2d, j and Supplementary Data 7 and 12). To further evaluate the specific effect of ADHD comorbidity within the ASD families, we compared ASD-M (ADHD+) versus ASD-M (ADHD−) and ASD-S (ADHD+) versus ASD-S (ADHD−), adjusting for age, gender, and dietary factors (Supplementary Fig. 2f, l). Notably, no significant bacterial species were identified in either comparison after multiple testing correction, suggesting that ADHD status alone does not yield significant microbiome differences within the same familial ASD subtype when controlling for other confounding factors.

These results suggest that the gut microbiome of children from ASD-M is more distinct from that of TD children than the microbiome of children from ASD-S. This indicates a more pronounced gut dysbiosis in the ASD-M than ASD-S. One possible explanation for these observations is that cohabitation with an affected sibling may exacerbate gut microbial alterations in ASD.

### The dissimilarity between ASD children and their cohabitation siblings within families
In this study, all the sibling pairs shared a household. Similar to previous reports about the widely shared microbiome among cohabitants[23,24], there were higher Bray–Curtis similarities at species levels in cohabiting sibling pairs from three multi-child families, when compared to unrelated random non-cohabiting pairs and random ASD pairs from ASD-O group (Kruskal–Wallis, $p = 2.6 \times 10^{-15}$, $\chi^2(4) = 74.449$, Fig. 2 and Supplementary Data 14). Moreover, among the cohabiting sibling pairs from the three types of families, simplex ASD sibling pairs and multiplex ASD sibling pairs showed higher Bray–Curtis similarities in bacterial species levels ($p = 0.022$ and $p = 0.13$, two-sided Wilcoxon rank-sum test, Fig. 2 and Supplementary Data 14) compared with TD sibling pairs. These findings further illustrated that ASD families were likely to have a more similar gut microbiome, indicating that ASD children may be more easily affected by their siblings than TD children.

### Microorganism strain-sharing between cohabitating siblings
Previous research has examined gut microbiome similarities among household members and strain transmission in cohabiting populations and more distant contacts[22]. Our cross-sectional study investigates

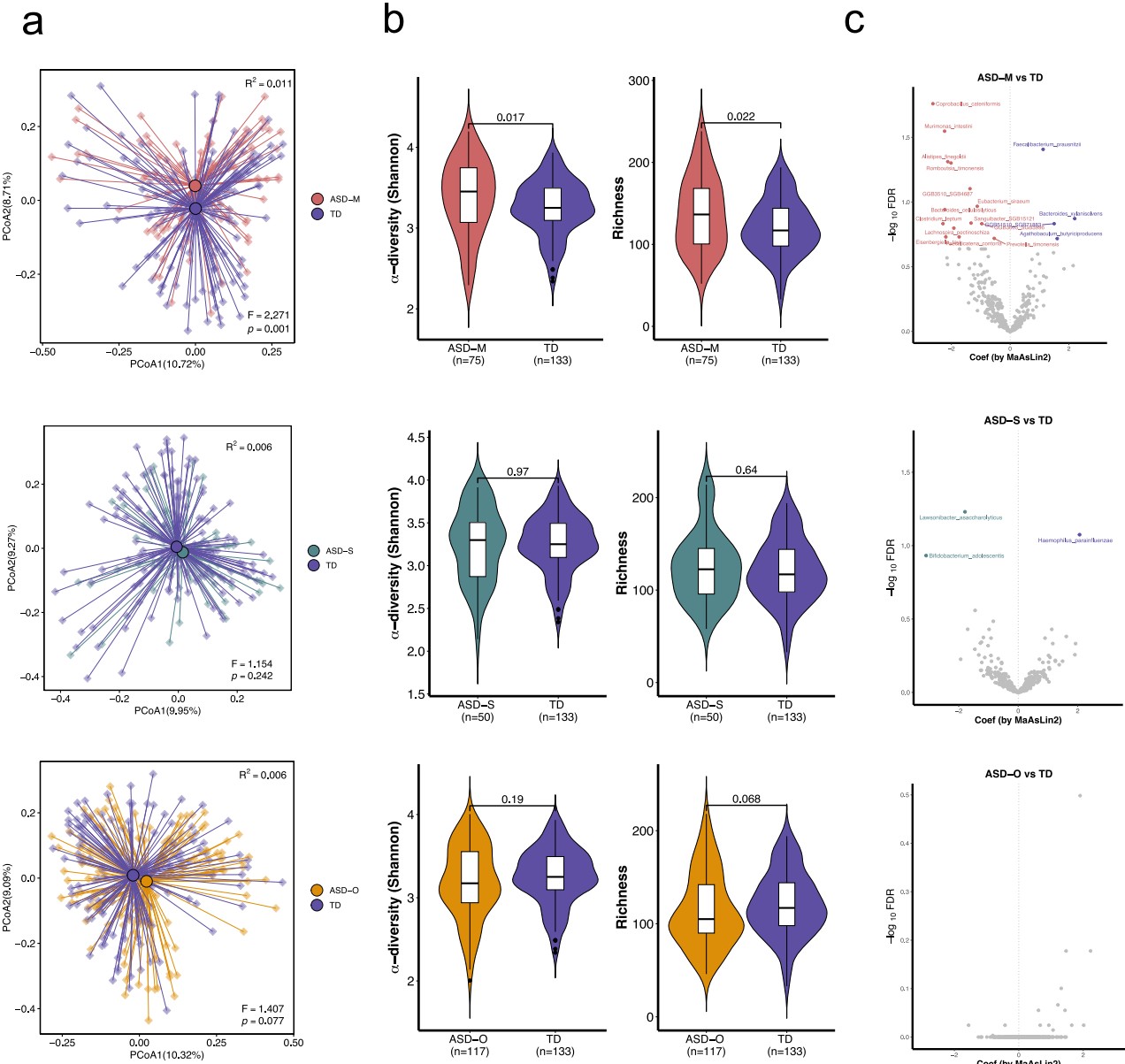

**Fig. 1 | Bacterial alterations in the gut microbiota of ASD from different family types. a** Principal coordinates analysis (PCoA) of gut microbiota composition based on Bray–Curtis dissimilarity, comparing ASD children from multiplex (ASD-M), simplex (ASD-S), and only-child (ASD-O) families with typically developing (TD) children. Group differences were assessed by PERMANOVA. **b** Diversity (Shannon Index) and richness of gut microbiota of ASD and TD children from different family types. In the box plots (embedded within violin plots), the center line represents the median; box bounds indicate the 25th and 75th percentiles; whiskers extend to the minima and maxima (within 1.5 times the interquartile range). Pairwise comparisons were performed using two-sided Wilcoxon rank-sum tests with Benjamini–Hochberg correction. **c** Differentially abundant bacterial species associated with each ASD family type compared to TD, identified using multivariate linear models (MaAsLin2; significance threshold: $p < 0.05$, FDR < 0.2).

strain sharing among sibling pairs in different family types. At the bacterial strain level, we observed significantly elevated strain-sharing rates in sibling pairs from both multiplex and simplex ASD families relative to TD families, with sibling pairs from multiplex families sharing significantly more strains than those from simplex families (Kruskal–Wallis test, $p = 0.00053$, $\chi^2(2) = 15.098$; Fig. 3 and Supplementary Data 15 and 16). Consistent with previous observations that microbiome transmission occurs beyond close contacts and reflects population structure, we observed distinct strain-sharing patterns across family types. This was evidenced by both the density distributions of sharing rates (Supplementary Fig. 4a and Supplementary Data 17) and a PERMANOVA performed on the unsupervised strain-sharing network (Euclidean distance matrix; $R^2 = 10.6\%$, $F = 44.08$,

$p = 0.001$; based on 999 permutations; Supplementary Fig. 4b and Methods). Network topology analysis revealed that the extensive distribution of Multiplex families in the visualization corresponds to a highly interconnected structure rather than a loose association, followed by simplex ASD families, and then TD families. Sibling pairs from multiplex ASD families exhibited significantly higher clustering coefficient compared to other families (Kruskal–Wallis test, $\chi^2(2) = 159.4$, $p < 2 \times 10^{-16}$; Supplementary Fig. 4c), suggesting the formation of tight-knit strain-sharing communities. These topological metrics confirm that strain transmission in multiplex ASD families is more robust and pervasive.

To evaluate gut microbiome sharing between siblings across different family types, we employed the chi-squared test with Yates'

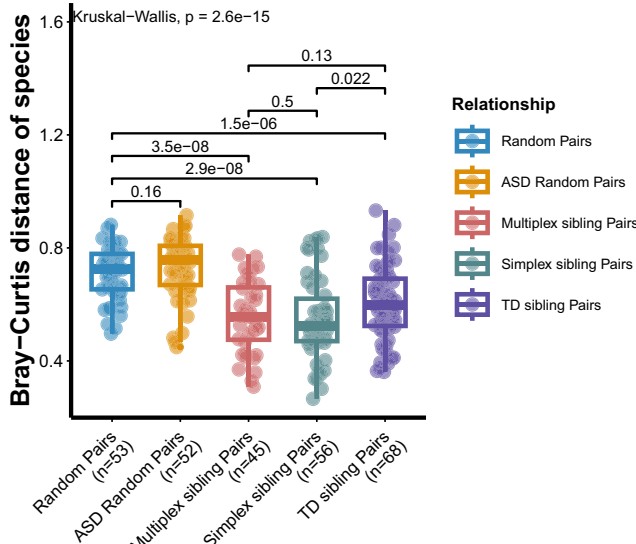

**Fig. 2 | Gut microbiome dissimilarity across cohabiting and non-cohabiting pairs.** Pairwise Bray–Curtis dissimilarity was compared among: random non-cohabiting pairs ($n = 53$), random ASD non-cohabiting pairs from ASD-O ($n = 52$), multiplex sibling pairs ($n = 45$), simplex sibling pairs ($n = 56$), and TD sibling pairs ($n = 68$). In the box plots, the center line represents the median; box limits indicate the 25th and 75th percentiles; whiskers extend to the maximum and minimum values. Kruskal–Wallis test showed significant differences among the five groups ($\chi^2(4) = 74.449$, $p = 2.6 \times 10^{-15}$). Post hoc pairwise comparisons were performed using two-sided Wilcoxon rank-sum tests with Benjamini–Hochberg correction.

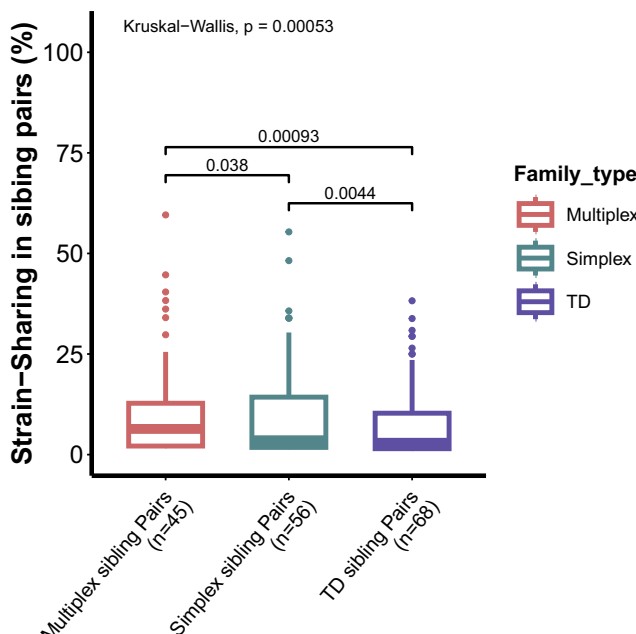

**Fig. 3 | Gut microbial strain-sharing rates across sibling pairs from different family types.** The strain-sharing rate was calculated as (number of shared strains/total sibling pairs) × 100%. Box plots showing strain-sharing rates in multiplex (ASD-M), simplex (ASD-S), and TD sibling pairs. In the box plots, the center line represents the median; box limits indicate the 25th and 75th percentiles; whiskers extend to the maximum and minimum values. A Kruskal–Wallis test indicated significant differences among groups ($\chi^2(2) = 15.098$, $p = 5.27 \times 10^{-4}$, $\varepsilon^2 = 0.038$, [95% CI: 0.005–0.085]). Post hoc pairwise comparisons were performed using two-sided Wilcoxon rank-sum tests with Benjamini–Hochberg correction. All pairwise differences were significant: multiplex vs. simplex, $p_{adj} = 0.0378$; multiplex vs. TD, $p_{adj} = 0.0028$; simplex vs. TD, $p_{adj} = 0.0066$. ε2: epsilon-squared effect size.

continuity correction or Fisher's exact test. Notably, several bacterial strains showed significantly higher sharing rates in multiplex ASD families compared to TD controls, including taxa with reported opportunistic or context-dependent pathogenic potential, such as *Eubacterium rectale* (SGB4933), *Dorea formicigenerans* (SGB4575), and *Acidaminococcus intestini* (SGB5736) ($p_{adj} < 0.05$, Fig. 4 and Supplementary Data 20). In contrast, the sharing of *Bacteroides xylanisolvens* (SGB1867) was markedly lower in multiplex ASD families than in the other two groups. Interestingly, we also observed an increased prevalence of sharing for several common gut commensals in ASD families. For instance, *Bifidobacterium pseudocatenulatum* (SGB17237) and *Faecalibacterium prausnitzii* (SGB15342) exhibited their highest sharing percentages in multiplex ASD families, while *Sellimonas intestinalis* (SGB4617) showed a significantly increased sharing rate specifically in simplex ASD families ($p_{adj} < 0.05$, Fig. 4 and Supplementary Data 20). These differential sharing patterns underscore a strong association between familial context and gut microbiome configuration in ASD. Rather than direct transmission, these findings indicate that children from ASD families exhibit distinct strain-sharing patterns compared with TD families. These patterns, which may be influenced by shared environmental and host factors, suggest that the gut microbiome configurations in ASD families are associated with the increased persistence or detectability of specific bacterial strains. This identifies the shared familial microbiome as a hypothesis-generating context that may be relevant for future interventional studies, pending longitudinal and mechanistic validation.

### Gut microbiome is associated with ASD clinical scores

To extend the findings and explore the clinical relevance. We processed the association analysis of the markers and clinical parameters. We found 7 multiplex ASD families enriched markers based on the MaAslin2 were significantly positively correlated with several ASD-related clinical parameters, including Social Responsiveness Scale

(SRS), Social Experience Questionnaire (SEQ), Child Behavior Checklist (CBCL), Anxiety scale for children-ASD (ASC-ASD) (Spearman's correlation, $p < 0.05$, Supplementary Fig. 5), including *Coprobacillus cateniformis*, *Murimonas intestini*, *Eisenbergiella tayi*, *Alistipes finegoldii*, *Lachnospira pectinoschiza*, *Sanguibacter SGB15121*, and *Eubacterium siraeum*. In contrast, the TD-enriched species *Faecalibacterium prausnitzii*, GGB51510_SGB71883, *Agathobaculum butyriciproducens*, and *Bacteroides xylanisolvens* were negatively correlated with the ASD clinical parameters (Spearman's correlation, $p < 0.05$, Supplementary Fig. 5).

## Discussion

Most human ASD gut microbiome studies have focused on the difference between ASD and TD[13,25,26]. However, they are rarely invested in the gut microbiome among different types of ASD families. In this study, we performed a comprehensive analysis of the metagenomic profiles of feacal samples from 429 children, the results showed that ASD-M have higher abundance of potential pathogens than ASD-S when compared to TD, respectively, highlighting more severe gut dysbiosis in ASD-M. We also find several beneficial bacteria that are negatively associated with ASD-M, such as *Faecalibacterium prausnitzii*[27,28], *Bacteroides xylanisolvens*[29,30], and *Agathobaculum butyriciproducens*[31]. Our correlation analysis also indicated that differential bacteria between ASD-M and TD are associated with ASD-related scores, thereby enhancing the reference value of our clinical observations.

Cohabitation has been associated with shared gut microbiome composition across family members[32–34], but rare studies assessed the effect of cohabitation on children with ASD. Our results revealed that

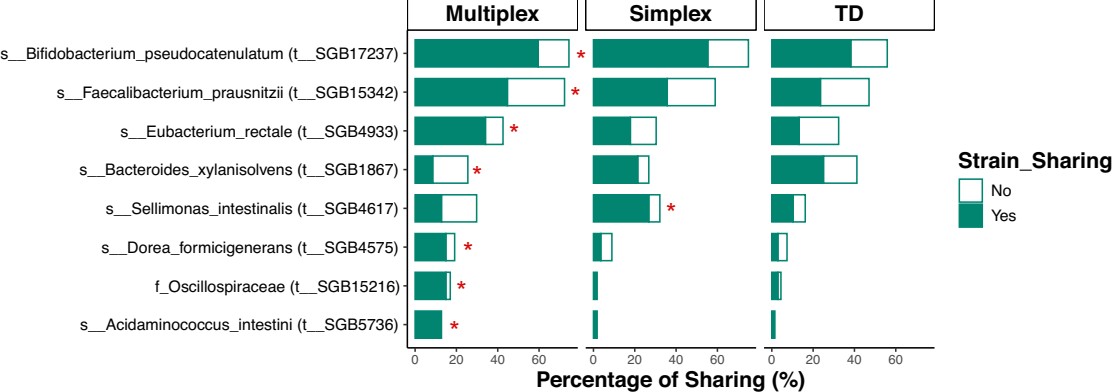

**Fig. 4 | Gut microbiome sharing between siblings among different family types.** All comparisons are statistically significant. Exact $p$ values and adjusted $p$ values ($p_{adj}$) for all comparisons are provided in Supplementary Data 20. Chi-squared test with Yates's correction, *$p_{adj} < 0.05$; Fisher's exact test, *$p < 0.05$.

ASD sibling pairs shared a more similar microbiome profile when compared to that of TD sibling pairs. These findings suggest that the presence of ASD children in a family may further enhance the similarity of the microbiome along with the cohabitation factors.

Several studies revealed the impact of cohabitation on the microbiome strain-sharing among individuals[21,22,35]. Our results at the strain level provide further evidence that ASD children, particularly those from multiplex families, shared more microorganisms with cohabitation siblings than TD children. This elevated sharing is characterized by a group of microorganisms typically recognized for their opportunistic or context-dependent pathogenic potential, including *Eubacterium rectale* (SGB4933), *Dorea formicigenerans* (SGB4575), and *Acidaminococcus intestini* (SGB5736). These findings indicated that children in multiplex ASD families may share the same susceptibility to potential pathogens, while children in simplex ASD families may not be associated with severe gut dysbiosis. These observations suggest that the high strain-sharing rate observed in multiplex ASD families may be associated with cohabitation or their shared environmental exposures, particularly given that our initial species-level findings demonstrated more severe gut dysbiosis and a higher abundance of opportunistic pathogens in ASD-M individuals. Furthermore, the differential bacteria identified between ASD-M and TD individuals in our results showed correlative trends with ASD-related clinical scores, highlighting a potential clinical relevance. While we interpret the apparent reduced resistance in their gut microbiota as an observational finding, it may also be linked to potential common genetic susceptibility factors. Moreover, the sharing rates of three potential beneficial bacterial strains—*Faecalibacterium prausnitzii* (SGB15342), *Bifidobacterium pseudocatenulatum* (SGB17237), and *Sellimonas intestinalis* (SGB4617)—were significantly higher in ASD families compared to TD families. This indicates that intestinal susceptibility may not solely present disadvantages. The characteristic ease of colonization and high environmental transmissibility of these specific strains highlight a potential avenue for intervention strategies from a different perspective. These results point out the potential of microbiome-based therapeutics in alleviating the symptoms of ASD. Targeting shared beneficial bacteria in ASD families might be an effective intervention approach because these bacteria might be easily absorbed or colonized in the intestines.

Although a combination of genetic vulnerability and environmental triggers likely plays a role in the development of ASD in some individuals[36], most of the genetic variants are associated with heterogeneous phenotypes[37]. Notably, large-scale genetic studies have focused on different ASD family types, suggesting that multiplex families have different genetic architectures from simplex families and may be more attributable to rare, dominant variants[7–12]. Besides, simplex families are more likely to have some rare de novo ASD risk variants, while multiplex families have shared genetic vulnerabilities[38]. On the other hand, previous studies indicated that host genetic variation can shape the gut microbiome[39,40]. Additional studies demonstrated the impact of host genetics on the diversity and abundance of the gut microbial taxa and even regulated the genetic diversity of gut microorganisms[40–45]. In the context of our study, host genetics were not directly assessed in the current progress. Therefore, we speculatively suggest that a shared genetic background in multiplex families might potentially contribute to the observed strain-sharing patterns and microbiome convergence, alongside shared environmental exposures. However, without corresponding host functional or genetic data, these potential links remain speculative. Further integrative studies are needed to elucidate the potential associations between host genetic backgrounds and the heterogeneity of the gut microbiome across different ASD family types.

As an observational cross-sectional study, our work has several limitations. First, although we have identified specific gut microbiome signatures in children with ASD across different family types, we currently lack sufficient data to elucidate the precise mechanisms driving this phenomenon. Crucially, no inference regarding the directionality of our findings can be made. As a cross-sectional study, our design cannot distinguish between direct microbial transmission, co-acquisition from shared environments, or microbiome convergence driven by unmeasured host factors or common genetic susceptibility; therefore, directionality and causality cannot be inferred. Second, while we adjusted for collected covariates, observational human microbiome studies are inherently subject to unmeasured confounders, such as specific household environmental factors (e.g., the number of pets) that were not recorded. Finally, as all participants in this study were Hong Kong Chinese, the generalizability of these findings to other ethnic or geographical populations remains to be established. Further longitudinal studies and mechanistic investigations are required to distinguish between these ecological processes and validate our findings in more diverse cohorts.

In conclusion, our observational microbiome study demonstrated new insights into the effect of family types on the gut microbiome of children with ASD and provides valuable and informative insights for future ASD research.

## Methods
### Ethics statement
The study was approved by the Joint Chinese University of Hong Kong of New Territories East Cluster Clinical Research Ethics Committee (CUHK-NTEC CREC). Written informed consent was obtained from the

parents or legal guardians of all participating children prior to their inclusion in the study.

## Study population

A cohort consisting of different family types was recruited from the Child and Adolescent Psychiatric Clinic of the New Territory East Cluster (NTEC) of the Hospital Authority and the community. In addition to a self-reported diagnosis of ASD, parents were required to provide a formal medical certificate from a qualified psychiatrist, pediatrician, or psychologist confirming the ASD diagnosis before their child was included in the case group for recruitment. Exclusion criteria included the use of probiotics, antidepressants, anti-epileptics, or antibiotics within the 1 month prior to enrollment. A total of 429 children were recruited in this study with phenotypic data (Supplementary Data 1) and diet. For the typically developing (TD) children, individuals with a positive screening on the Autism Spectrum Quotient-10 (AQ-10) will be excluded from the recruitment. Among the 429 children, 312 of them were from three types of multi-child families, and the other 117 children were from only-child ASD families. Among the multi-child families, the "multiplex" ASD families were defined as families with two or more children with a confirmed diagnosis of ASD, "simplex" families were defined as families with only one child diagnosed with ASD and at least one unaffected male sibling, and "TD" families were defined as families without children with ASD.

## Stool sample collection

Fecal samples were collected at home by all subjects using stool specimen collection tubes prepared by investigators with preservative media (cat. 63700, Norgen Biotek Corp, Ontario, Canada). The Norgen preservative can effectively preserve microbial DNA and RNA during transportation at room temperature, ensuring sample integrity and eliminating variability. Stool samples were delivered to the hospital within 24 h after collection and stored at −80 °C refrigerators until further processing. In our previous study, we have demonstrated that the composition of gut microbiota data obtained from fecal samples collected using this preservative medium is comparable to data derived from fresh samples that were promptly stored at −80 °C[46]. Also, the storage duration exhibited comparability between cases and controls across all individuals involved in the study.

## Stool DNA extraction and sequencing

All fecal samples from this study were handled in a random sequence according to the same protocol to avoid potential batch effects. In brief, following the removal of the preservative media, microbial DNA isolation was performed using the Qiagen (Hilden, Germany) DNeasy PowerSoil Pro Kit, under the instructions provided by the manufacturer. After the quality control procedures by Qubit 2.0, agarose gel electrophoresis, and Agilent 2100, the extracted DNA was utilized for constructing DNA libraries. Library construction involved end repairing, the addition of A to tails, purification, and PCR amplification, using the Illumina® DNA Prep with (M) Tagmentation kit (Illumina, San Diego, CA). Libraries were subsequently sequenced on our in-house Illumina NovaSeq (150 base pairs paired-end). To ensure the reliability of the results, ZymoBIOMICS Microbial Community Standard (Cat: D6300, ZYMO Research, USA) and ZymoBIOMICS Microbial Community DNA Standard (Cat: D6306) were used as positive controls throughout the DNA extraction, library construction, sequencing, and quality assessment processes. In cases where abnormal signals were detected, resequencing was performed to ensure data accuracy.

## Sequencing data preprocessing

The raw sequence data were quality filtered using Trimmomatic (v39) to eliminate adapter sequences, low-quality reads (quality score <20), and reads shorter than 50 base pairs[47]. The remaining reads were mapped to the mammalian genome (hg38, felCat8, canFam3, mm10, rn6, susScr3, galGal4, and bosTau8; UCSC Genome Browser), bacterial plasmids (accessed from the National Center for Biotechnology Information (NCBI) RefSeq database in January 2023), complete plastomes (accessed from the NCBI RefSeq database in January 2023), and UniVec sequences (accessed from the NCBI RefSeq database in January 2023) using bowtie2 v.2.4.2[48]. Reads that potentially originated from the host or laboratory-associated sequences were removed as contaminant reads using KneadData v.0.6. To expedite data processing, GNU parallel was employed for parallel analysis tasks.

Shotgun metagenomic sequencing was performed on the Illumina platform for all 429 stool samples with an average depth of 6 Gb. While the yield varied across samples, we generated a total of approximately 3.6 Tb of raw data, with a targeted coverage of approximately 2X (assuming a baseline of 3 Gb for 1X coverage of the human gut metagenome). Although sequencing yield varied across samples, our downstream analytical pipeline was designed to ensure high-resolution profiling across this range.

## Microbial taxonomic profiles

Microbial composition profiles were analyzed from the quality-filtered forward reads using MetaPhlAn4[49] (version 4.1.4) with default parameters and the vOct22_202212 database. To accelerate processing, all analyses were run in parallel using GNU parallel (version 2018). Taxonomic profiling at the strain level was performed using StrainPhlAn4[50] (version 4.1.4) with default parameters.

Then, the strain-level profiling was performed with StrainPhlAn4 using the custom species-level genome bins (SGBs) marker database with the following parameters: " marker_in_n_samples_perc 50 -sample_with_n_markers 10 -phylophlan_mode accurate -mutation_rates". The SGBs detected with "-print_clade_only" were selected for strain-level profiling. This marker-gene-based approach has been benchmarked to provide robust strain-level identification even at an average coverage of 1.3× per SGB, consistently outperforming assembly-based methods in samples with moderate sequencing depth.

## Detection of a strain-sharing event

Strain sharing was assessed by profiling each sample, with the dominant strain of SGBs with a set of SGB-specific normalized phylogenetic distance (nGD) thresholds in each strain by a previous study from five published metagenomic datasets in four countries[22,51–55]. nGDs were computed as the ratio of leaf-to-leaf branch length normalized by the total branch length of the phylogenetic trees generated by StrainPhlAn. These trees were constructed based on the marker gene alignments, specifically targeting positions with a minimum variability of 1%. For the SGB-specific nGDs that were not estimated previously, the nGD value corresponding to the 3rd percentile of the distribution of nGD values among individuals was used. To enhance the reliability of strain identity threshold selection, we derived an alternative threshold for the 105 SGBs with n_related ≥5 (i.e., detected in at least 5 sibling pairs). The species-specific strain identity threshold was calculated based on Youden's index and compared against the threshold obtained from the method described earlier; the more conservative of the two was selected as the final strain identity threshold (Supplementary Data 18). For the remaining strains with insufficient sharing events, we retained the SGB-specific nGD thresholds derived from previous studies or the 3rd percentile of the nGD distribution across individuals.

In detail, to evaluate the reproducibility of the species-specific strain-identity thresholds within our dataset, we included all 429 samples from related sibling pairs and unrelated subjects. This validation assessed the separation between the distributions of nGD distances for strains from the same SGB in these two groups. The analysis was performed for the 105 SGBs detected in at least 5 sibling pairs in this cohort. Given the absence of longitudinal samples, we defined true positives as pairwise phylogenetic distances (nGD) between sibling

pairs that fell below the species-specific strain identity threshold—obtained either (1) from independent longitudinal datasets of previous studies or as the 3rd percentile of the inter-individual nGD distribution, or (2) computed using Youden's index from our dataset. False positives were defined as nGD values from balanced unrelated pairs below the threshold, true negatives as those from balanced unrelated pairs above the threshold, and false negatives as nGD values from sibling pairs above the threshold. Performance metrics for the two thresholds in distinguishing strains from sibling pairs versus unrelated pairs were as follows: median precision = 0.97 and 0.98, and median $F$-score = 0.72 and 0.77 (Supplementary Fig. 3 and Supplementary Data 19).

### Strain–sharing networks

Unsupervised strain-sharing networks were constructed and visualized using the R packages tidygraph (v1.2.3), igraph (v1.4.3), and plotted with ggraph (v2.1.0). Networks were laid out using a stress-minimization algorithm. Nodes represent individuals, and edges connect pairs that share ≥10 bacterial strains. Only edges meeting this threshold are displayed. We calculated the Local Clustering Coefficient for each node using the igraph package in R.

### Statistics and reproducibility

No statistical method was used to predetermine sample size. The sample size ($n = 429$) was determined by the available recruitment from the Child and Adolescent Psychiatric Clinic and the community during the study period. No data were excluded from the analyses after the recruitment phase based on the predefined exclusion criteria (e.g., use of antibiotics or probiotics within 1 month prior to enrollment). The experiments were not randomized; subjects were categorized into groups (Multiplex, Simplex, TD families, and ASD-O group) based on their clinical diagnosis and family structure. The Investigators were not blinded to allocation during experiments and outcome assessment, although all fecal samples were processed in a random sequence using standardized protocols to minimize potential batch effects.

The reproducibility of the metagenomic pipeline was ensured through the inclusion of internal positive controls (ZymoBIOMICS Microbial Community Standard and DNA Standard) throughout DNA extraction, library construction, and sequencing. In cases of abnormal signals, resequencing was performed to ensure data integrity. The robustness of the strain-level profiling was validated by benchmarking against unrelated subject pairs, achieving a median precision of 0.97 and a median $F$-score of 0.72. Statistical analyses, including the calculation of local clustering coefficients and network visualizations using stress-minimization algorithms, were performed using R with the tidygraph, igraph, and ggraph packages.

## Data availability

The metagenomic sequencing data generated in this study have been deposited in the NCBI Sequence Read Archive database under accession code PRJNA1377943. The results of the analyses, including all exact $p$ values, are available in the Supplementary Data. The corresponding legends can be found in the Description of Additional Supplementary Files. Participant metadata cannot be made publicly available via repositories as outlined in the patient consent form to protect participant privacy. Requests for sharing metadata, including dietary profiles, can be submitted with a written proposal to the corresponding author (S.C.N.) at siewchienng@cuhk.edu.hk. The proposal should detail the intended use of the data. The data management team, composed of scientists and clinicians, will review these requests based on scientific merit and ethical considerations, including patient consent, to avoid any misuse or misinterpretation. Data sharing will be undertaken if the proposed projects have a sound scientific rationale or potential patient benefit. Data recipients are required to enter a formal data sharing agreement, which describes the conditions for release and requirements for data transfer, storage, archiving, and publication. Since the data management meeting is held monthly, please anticipate a response within 2 working months. Data access is typically granted for 12 months under a Data Use Agreement that prohibits participant re-identification and third-party data transfer.

## Code availability

All software used were sourced from publicly available repositories. The codes utilized for the microbiome analyses or figures can be accessed at the Zenodo repository (https://doi.org/10.5281/zenodo.18505267). The MetaPhlAn4 package[49] was used for the species- and strain-level analyses, available at the GitHub repository (https://github.com/biobakery/MetaPhlAn).

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

## Acknowledgements

We appreciate the assistance from Ms. Man Ki Fong for the participants' enrollment. We would like to thank Ms. Crystal Wong and Ms. Uuriinsaran Purevsuren for assistance with DNA extraction and sequencing. This study was supported by InnoHK, the Government of Hong Kong, Special Administrative Region of the People's Republic of China, The D. H. Chen Foundation, the New Cornerstone Science Foundation through the New Cornerstone Investigator Program, the Hong Kong Jockey Club Charities Trust, the National Natural Science Foundation of China (82400646) and the Research Grants Council of the Hong Kong Special Administrative Region, China (CUHK14104124).

## Author contributions

W.L. conceived the study, performed metagenomic sequencing, ran analyses, and took responsibility for the preparation of the manuscript. O.W. worked with W.L. on primary analysis and performed a clinical assessment. J.Z. and S.Chen. contributed to part of the data analyses. Y.W. and Z.X. contributed to metagenomic sequencing. C.P.C., J.Y.C., and P.K.C. contributed to participant recruitment, sample collection, and biobank management. S.Chan., S.W., and D.C. helped initiate the study and contributed to the participant recruitment. H.M.T. and F.K.L.C. contributed to the study design and data interpretation. Q.S. and S.C.N. oversaw the entire study and contributed to the study design, interpretation, and manuscript writing. All authors gave final approval for the version to be published.

## Competing interests

F.K.L.C. is a Board Member of CUHK Medical Centre. He is a co-founder, non-executive Board Chairman, honorary Chief Medical Officer, and shareholder of GenieBiome Ltd. He receives patent royalties through his affiliated institutions. He has received fees as an advisor and honoraria as a speaker for Eisai Co. Ltd., AstraZeneca, Pfizer Inc., Takeda Pharmaceutical Co., and Takeda (China) Holdings. Co. Ltd. S.C.N. has served as an advisory board member for Pfizer, Ferring, Janssen and Abbvie and

received honoraria as a speaker for Ferring, Tillotts, Menarini, Janssen, Abbvie and Takeda; has received research grants through her affiliated institutions from Olympus, Ferring and Abbvie; is a founder member, non-executive director, non-executive scientific advisor and shareholder of GenieBiome Ltd which is non-remunerative; is a shareholder of MicroSigX Diagnostic Holding Limited; is a founder member, non-executive Board Director, and non-executive scientific advisor of MicroSigX Biotech Diagnostic Limited, which is non-remunerative; and receives patent royalties through her affiliated institutions. Q.S., S.C.N., and F.K.L.C. are named inventors of patent applications held by the CUHK and MagIC that cover the therapeutic and diagnostic use of the microbiome. The remaining authors declare no competing interests.
