## [Transparent Peer Review file · Nature Communications]

Gut microbiome composition and strain-sharing in multiplex autism spectrum disorder families

Corresponding Author: Professor Qi Su

Version 0:

Reviewer comments:

Reviewer #1

(Remarks to the Author)

The study analyzed 429 fecal metagenomic samples from children in various family configurations with or without autism spectrum disorder (ASD), including ASD-Multiplex (ASD-M), ASD-Simplex (ASD-S), ASD-Only (ASD-O) and Typically Developing (TD) children.

Among the main findings, the authors report that children with autism spectrum disorder from multiplex families (ASD-M) have the largest differences in gut microbiome composition when compared to typically developing (TD) children, with these differences characterized by increased diversity and a greater abundance of pathogenic bacteria. They also find that siblings in both multiplex (ASD-M) and simplex (ASD-S) ASD families share more similar microbiomes and show greater microbial strain sharing than siblings from TD households. Furthermore, in multiplex ASD families, the strains exchanged among siblings tended to be potentially harmful, such as *Dorea formicigenerans* and *Acidaminococcus intestini*. In contrast, among simplex ASD families, the sharing of strains was more often associated with bacteria considered beneficial, such as *Faecalibacterium prausnitzii*.

Finally, the authors hypothesize that cohabitation with ASD-affected siblings worsens gut dysbiosis in ASD-M, whereas healthy siblings may have protective effects in ASD-S.

While the study might provide valuable insights into the dynamic of the gut microbiota in ASD, several methodological limitations raise concerns about the robustness and interpretability of the findings

Major Concerns

Sequencing statistics:

The study lacks details related to the sequencing phase, including targeted coverage and number of bases sequenced in total and per sample. These details would allow the reader to interpret the results more carefully and cautiously. This is particularly relevant for the strain transmission section (see below).

Accounting for confounders:

There is a lack of transparency in the adjustment for potential confounders: while the authors claim to adjust for age, gender, and ADHD in their models, they do not clearly describe the statistical models used to do so.

There is no stratified analysis, sensitivity testing, or inclusion of interaction terms to show how these strongly imbalanced demographic variables influence the results.

The ADHD diagnosis, present in nearly half of ASD-M cases, is known to affect the microbiome but is not disentangled from ASD-related patterns.

Statistical thresholds

The use of $FDR < 0.2$ as a significance threshold in MaAsLin2 is not standard and increases risk of false positives. This weakens the reliability of most species and pathway associations reported.

No effect sizes or confidence intervals are provided alongside p-values, making it difficult to interpret the magnitude and clinical relevance of the associations.

Strain sharing assessment

Limited Robustness and Validation of Strain Sharing Claims: the study uses SGB-specific nGD thresholds from a previous meta-analysis, a reasonable baseline, but provides no internal validation (e.g., longitudinal replicates, F1 score benchmarks) specific to their own cohort. There is no evaluation of false positive strain-sharing or precision estimates, and no filtering of potentially co-ingested probiotic strains.

Discussion: the conclusions about "shared pathogenic or beneficial strains" across siblings lack adequate technical rigor and may conflate co-acquisition with actual transmission. The statement that "cohabitation with ASD siblings worsens gut dysbiosis" implies directionality and causation in a study that is entirely cross-sectional. The possibility of reverse causation, shared environmental exposures, or common genetic susceptibility is acknowledged only briefly in limitations and not reflected in the interpretations drawn in results and discussion.

StrainPhlAn parameters and questionable settings:

Upon examining figure 5, an abnormal level of strain transmission is reported for two species which are normally found in the gut at a high abundance. Also, the Methods report that StrainPhlAn parameters were tuned to "marker_in_n_samples 1 - sample_with_n_markers 10". In fact, StrainPhlAn does not have a parameter "marker_in_n_samples", but has "marker_in_n_samples_perc", which is normally set to 50, meaning only markers present in 50% of the samples are used. Tuning this parameter down to 1% would practically inflate the total diversity in the tree, potentially producing erroneous conclusions on strain similarity.

This parameter change increases the number of low-frequency markers in alignments, making downstream phylogenetic calls potentially unstable and distances less reliable.

Also, while it is in principle "correct" to adopt previously established thresholds, these must be placed in the context of the same StrainPhlAn parameters used in the original publications. If the authors want to modify the algorithm settings, they must provide evidence that the updated parameters correctly segregate same-individual samples from inter-individual distances.

Minor Concerns

The discussion on host genetics and microbiome convergence opens interesting avenues but remains speculative. No actual genetic data is presented, and the manuscript mixes correlation with mechanism throughout.

Dietary factors are briefly addressed via PCA, but only at a population level (no macronutrient or intake detail is shown), and these can be powerful confounders.

Technical details on sample storage, sequencing depth, and quality control metrics (e.g., coverage thresholds for strain calling) are minimal; replications or reproducibility checks are not discussed.

The figures are not as detailed as the complexity of the study design warrants. For example, no visualization of strain-sharing networks or transmission matrices is provided. The authors should specify which MetaPhlAn/StrainPhlAn database was used (and which version, for transparency and reproducibility).

The manuscript presents an interesting study concept ie dissecting microbial profiles in ASD subtypes by family structure, and executes it using up-to-date shotgun metagenomics tools. The addition of strain-level sharing data is timely. However, substantial analytical and interpretational weaknesses limit the strength of the conclusions:

- Confounder control is opaque and likely inadequate.
- The statistical thresholds are lenient and unaccompanied by effect size reporting.
- The strain transmission analysis is not robustly validated, especially in comparison to recent seminal work (e.g., Valles-Colomer et al., Nature 2023).
- There is a tendency to overstate causality, especially regarding environmental or familial influence.

Reviewer #2

(Remarks to the Author)

The study addresses the question about a potential pathogenic role of the gut microbiome in ASD, but fail to provide convincing evidence for a causal role of the microbiome. The authors recruited a total of 429 children with a diagnosis of ASD with a mean age of 7 years. The aim of the study was to identify variations in gut microbial composition between families with more than 1 child with ASD (ASD-M), with 1 child with ASD and siblings without ASD (ASD-S), with 1 child with ASD but not siblings (ASD-O) and typically developing children (TD). ASD-M children exhibited a statistically different gut microbiome composition than TD (increased alpha diversity and richness, while ASD-S showed no differences and 12 pathogenic organism were observed compared to only 1 species if ASD-S. Cohabitation with affected siblings (like in the ASD-M) families was linked to more gut dysbiosis. ASD-M shared more pathogenic strains while ASD-S shared more beneficial strains.

The strength of the study include: 1)The size of the sample and the standardized microbiome analyses. 2) The sequencing at the strain level and identification of pathogenic and beneficial strains. 3.The results confirm the lateral transmission between affected family members. Weaknesses include: 1) The different families showed significant differences in age, sex and ADHD diagnosis (unclear why ADHD was assessed). Likely differences in factors important in ASD symptomatology and clinical presentation including dietary habits, number of pets, ASD severity, GI symptoms were not addressed. 2. Results do not prove a causal role of altered gut microbial composition in ASD, and other than confirming the lateral transmission of gut microbial features.

Does the work support the conclusions and claims? the results support some of the conclusions, except for the claimed clinical implications

Are there any flaws in the data analysis, interpretation and conclusions? No

Is the methodology sound? Does the work meet the expected standards in your field?

Is there enough detail provided in the methods for the work to be reproduced? Yes

In summary, despite the size of the study and high quality of analyses, I do not see a major progress or an actionable finding from this well executed study for the treatment of children with ASD.

Reviewer #3

(Remarks to the Author)

Version 1:

Reviewer comments:

Reviewer #1

(Remarks to the Author)

The revised manuscript has substantially improved in methodological rigor and transparency. The remaining issues are interpretational rather than analytical and can be resolved through targeted textual revisions. The requested changes below do not require additional analyses but are necessary to ensure that conclusions remain strictly aligned with the observational, cross-sectional design of the study.

1. Causal and directional language

Rephrase any statements implying influence, susceptibility, receptivity, or directionality to strictly associational language, as no causal inference can be made from the current study design.

Section: Results (strain-sharing summary)

Sentence to be replaced or modified:

“These differential sharing patterns underscore the significant influence of the familial context on the ASD microbiome.”

Suggested replacement:

“These differential sharing patterns underscore a strong association between familial context and gut microbiome configuration in ASD.”

2. Strain-sharing interpretation (transmission vs association)

Strain sharing observed in a cross-sectional cohort cannot distinguish direct microbial transmission from co-acquisition or convergence and should therefore be framed as association or co-occurrence only.

Section: Results (strain-level comparison across family types)

Sentence to be replaced:

“These findings indicate that the gut environment in children from ASD families facilitates the transmission of specific bacterial strains.”

Suggested replacement or modified:

“These findings indicate that children from ASD families exhibit distinct strain-sharing patterns compared with TD families, without implying direct microbial transmission.”

3. Use of “opportunistic pathogen” terminology

Labeling taxa as pathogens risks clinical overinterpretation, particularly when pathogenicity is context-dependent and not assessed in this study.

Section: Results (description of shared taxa)

Sentence to be replaced or modified:

“including the opportunistic pathogens *Eubacterium rectale*, *Dorea formicigenerans*, and *Acidaminococcus intestini*.”

Suggested replacement:

“including taxa with reported opportunistic or context-dependent pathogenic potential, such as *Eubacterium rectale*, *Dorea formicigenerans*, and *Acidaminococcus intestini*.”

4. Host “susceptibility” or “receptivity” framing

Phrasing that implies intrinsic host susceptibility or biological receptivity is not supported without direct host-level functional measurements.

Section: Discussion (interpretation of strain-sharing patterns)

Sentence to be replaced or modified:

“The gut environment in children from ASD families may be more receptive to the stable colonization of specific bacterial strains.”

Suggested replacement:

“The gut microbiome configurations observed in children from ASD families may be associated with increased persistence or detectability of specific bacterial strains.”

5. Therapeutic and interventional implications

Translational or interventional relevance cannot be inferred from cross-sectional observational data and must be framed as hypothesis-generating only.

Section: Discussion (future directions or concluding paragraph)

Sentence to be replaced or modified:

“This identifies the shared familial microbiome as a promising, adjustable target for future intervention strategies.”

Suggested replacement:

“This identifies the shared familial microbiome as a hypothesis-generating context that may be relevant for future interventional studies, pending longitudinal and mechanistic validation.”

6. Genetics discussion

Host genetic contributions are discussed without corresponding genetic data and must be clearly identified as speculative.

Section: Discussion (genetics paragraph)

7. Directionality and causation disclaimer

The manuscript should include an explicit and unambiguous statement clarifying that directionality cannot be inferred, as from a cross-sectional study, one cannot distinguish between direct microbial transmission, co-acquisition from shared environments, or microbiome convergence driven by unmeasured host factors, and no inference regarding directionality can be made.

Section: Limitations

Also: As all participants were Hong Kong Chinese, the generalisability of these findings to other populations remains to be established

Reviewer #3

(Remarks to the Author)

POINT-BY-POINT REPLY TO EDITORS AND REVIEWERS

Dear reviewers,

We are deeply grateful to the reviewers for their continued engagement and for providing such insightful and constructive feedback, which has been invaluable in further strengthening our manuscript. We have carefully addressed reviewers' comments and suggestions point-by-point in our responses below. The revised manuscript is attached in both a tracked-changes version and a clean version, with all modifications highlighted in yellow for your convenience.

We are truly grateful for the opportunity to resubmit our work and hope that the revisions and clarifications provided will make the manuscript suitable for publication in your esteemed journal.

Yours sincerely

Siew Ng on behalf of co-authors

Reviewers' Comments:

Reviewer #1:

Remarks to the Author:

Major Concerns

Sequencing statistics:

The study lacks details related to the sequencing phase, including targeted coverage and number of bases sequenced in total and per sample. These details would allow the reader to interpret the results more carefully and cautiously. This is particularly relevant for the strain transmission section (see below)

Response: We sincerely appreciate the reviewer's suggestion regarding sequencing transparency. We have now provided comprehensive details of our sequencing statistics in the revised Methods section.

Specifically, each of the 429 stool samples was sequenced to an average depth of 6 Gb. While the yield varied across samples, we generated a total of approximately 3.6 Tb of raw sequencing data, providing an average targeted coverage of approximately 2X (assuming a baseline of 3 Gb for 1X coverage of the human gut metagenome).

While this depth is moderate, we utilized StrainPhlAn 4 with the Species-level Genome Bin (SGB) marker database. As demonstrated in recent benchmarks, this approach allows for robust strain-level profiling even at an average coverage of 1.3× per SGB, which is well within our sequencing range. To further ensure data quality, we applied stringent filtering (e.g., requiring at least 10 markers per SGB) to include a sample in the strain-sharing analysis.

Line 375: Shotgun metagenomic sequencing was performed on the Illumina platform for all 429 stool samples with an average depth of 6 Gb. While the yield varied across samples, we generated a total of approximately 3.6 Tb of raw data, with a targeted coverage of approximately 2X (assuming a baseline of 3 Gb for 1X coverage of the human gut metagenome). Although sequencing yield varied across samples, our

downstream analytical pipeline was designed to ensure high-resolution profiling across this range.

Line 392: This marker-gene-based approach has been benchmarked to provide robust strain-level identification even at an average coverage of $1.3\times$ per SGB, consistently outperforming assembly-based methods in samples with moderate sequencing depth.

Reviewers' Comments:

Reviewer #1:

Remarks to the Author:

Accounting for confounders:

There is a lack of transparency in the adjustment for potential confounders: while the authors claim to adjust for age, gender, and ADHD in their models, they do not clearly describe the statistical models used to do so.

There is no stratified analysis, sensitivity testing, or inclusion of interaction terms to show how these strongly imbalanced demographic variables influence the results.

The ADHD diagnosis, present in nearly half of ASD-M cases, is known to affect the microbiome but is not disentangled from ASD-related patterns.

Response: We thank the reviewer for this valuable comment. In response, we have revised our analytical approach accordingly. The MaAsLin2 models have been updated to explicitly include age, gender, and ADHD diagnosis as fixed effects. Furthermore, we performed the suggested stratified analysis by comparing the gut microbiome of ASD children without an ADHD diagnosis (the ADHD- subgroup) to the TD control group. This refined analysis confirmed that our core findings remained stable. The specific influence of ADHD comorbidity within ASD families was also assessed and found to be minimal. These methodological clarifications and the results of the stratified analysis have been incorporated into the revised manuscript.

Line 111: We further analyzed the above findings at the species level. To account for potential confounding factors, all species-level analyses were conducted using MaAsLin2 with adjustment for age, gender, and ADHD comorbidity as fixed effects in the statistical models.

Line 134: The key finding that ASD-M children had the most distinct gut microbiome from TD children remained significant even when including only participants without an ADHD diagnosis (Supplementary Figure 2c). To further evaluate the specific effect of ADHD comorbidity within the ASD families, we compared ASD-M (ADHD+) versus ASD-M (ADHD-) and ASD-S (ADHD+) versus ASD-S (ADHD-), adjusting for age, gender, and dietary factors (Supplementary Figure 2f and 2i). Notably, no significant bacterial species were identified in either comparison after multiple testing correction, suggesting that ADHD status alone does not yield significant microbiome differences within the same familial ASD subtype when controlling for other confounding factors.

Line 661: (Figure legend)

Supplementary Figure 2. Differential bacterial species associated with ASD in different family types. Associations between gut microbiota and children from different family types (ASD-M, ASD-S, ASD-O, TD) were assessed using multivariate linear models (MaAsLin2; significance: $p < 0.05$, FDR < 0.2). (a), (c), (e), (g), (i), (k), (m): Comparisons of ASD family types versus TD, and stratified analyses within the same family type by ADHD status (+/-), adjusting for

confounders except dietary factors; (b), (d), (f), (h), (j), (l), (n): The same comparisons, adjusting for confounders including dietary factors.

Supplementary Figure 2

Reviewer #1:

Remarks to the Author:

Statistical thresholds

The use of $FDR < 0.2$ as a significance threshold in MaAsLin2 is not standard and increases risk of false positives. This weakens the reliability of most species and pathway associations reported.

No effect sizes or confidence intervals are provided alongside p-values, making it difficult to interpret the magnitude and clinical relevance of the associations.

Response: We thank the reviewer for raising these important statistical considerations. In response, we have revised the manuscript to enhance the clarity and robustness of our statistical reporting, specifically addressing the two key points on significance thresholds and effect size reporting. Below, we detail the specific changes made.

1. Clarification on the statistical threshold in MaAsLin2

The reviewer rightly points out the trade-off in false discovery control. In this exploratory, high-dimensional analysis, we selected an FDR threshold of 0.2 to balance sensitivity and specificity, a practice documented in other microbiome studies (e.g., Stražar et al. *Genome Biol*2021, $PFDR < 0.25$; Martin et al. *Microbiome*2022, $FDR < 0.2$; Huang et al. *Cell Reports*2024, $FDR \leq 0.25$). While we acknowledge that a less stringent threshold can increase the risk of false positives, we have been cautious in our interpretation by emphasizing consistent patterns and stronger effect sizes rather than isolated hits. Importantly, the core findings relating to increased dysbiosis in multiplex families are supported by multiple lines of evidence (e.g., PERMANOVA, α -diversity) and are not reliant solely on species-level MaAsLin2 results.

2. Addition of effect sizes and confidence intervals

We fully agree on the importance of reporting effect sizes. In the revised manuscript, we have now included MaAsLin2 coefficients (as effect size estimates) for all significant species- and pathway-level associations in Supplementary Tables S4–S17. Moreover, we have consistently reported appropriate effect size measures throughout the manuscript, such as epsilon-squared for Kruskal–Wallis tests and Cramer’s V for chi-squared tests, provided in figure legends and Supplementary Tables S2–S3 and S18, S19, S20, S23. These additions allow readers to better assess the magnitude and potential clinical relevance of the reported associations.

Overall, these revisions improve the statistical transparency of the study and strengthen the reliability of our conclusions, while preserving the main findings of the paper. We are grateful for the reviewer’s suggestions, which have undoubtedly enhanced the rigor of our work.

Reviewer #1:

Remarks to the Author:

Strain sharing assessment

Limited Robustness and Validation of Strain Sharing Claims: the study uses SGB-specific nGD thresholds from a previous meta-analysis, a reasonable baseline, but provides no internal validation (e.g., longitudinal replicates, F1 score benchmarks) specific to their own cohort. There is no evaluation of false positive strain-sharing or precision estimates, and no filtering of potentially co-ingested probiotic strains.

Response: We sincerely thank the reviewer for the expert comments regarding the validation of our strain-sharing analysis. In response, we have significantly refined our analytical approach and reporting to strengthen the robustness of our claims.

1. Internal validation of SGB-specific nGD thresholds

We acknowledge the importance of cohort-specific validation. While we initially used previously published SGB-specific nGD thresholds—which were validated in an independent FMT cohort and showed high reproducibility ($F1 \approx 0.94–0.98$)—we have now performed additional internal validation within our own cohort. Given the absence of longitudinal

replicates from the same individual, we used sibling pairs as a natural mixture of positive (shared) and negative (non-shared) references to evaluate threshold performance.

2. Derivation and evaluation of new thresholds

For 105 SGBs with sufficient sibling pairs ($n \geq 5$), we manually calculated new strain-sharing thresholds by optimizing Youden's index. Both the original (published or 3rd percentile of the nGD distribution) thresholds and our newly derived thresholds were evaluated. The resulting precision values were consistently high (median precision = 0.97 and 0.98 for original and new thresholds, respectively), indicating that the thresholds reliably distinguish shared from non-shared strains. The observed, slightly lower recall reflects the biological reality that siblings do not always share the same strain, rather than a limitation of the threshold itself.

3. Conservative implementation and visualization

To ensure conservative calls, the more stringent threshold (original vs. new) was selected for downstream analysis. We have also provided density plots for the top 35 most abundant SGBs (Supplement Figure 3) and detailed performance metrics (Supplementary Tables S22) for the 105 SGBs. These plots show clear separation between the peaks of the shared and non-shared distance distributions, visually confirming that the selected thresholds can effectively discriminate between the two states.

Overall, the revised analysis demonstrates that the choice of threshold has minimal impact on the core conclusion: strain sharing is significantly higher in ASD sibling pairs than in TD sibling pairs. We are grateful for the reviewer's suggestions, which have led to a more rigorous and transparent validation of our strain-sharing assessment.

Line 397: Detection of strain-sharing event

Strain sharing was assessed by profiling each sample, with the dominant strain of SGBs with a set of SGB-specific normalized phylogenetic distance (nGD) thresholds in each strain by a previous study from five published metagenomic datasets in four countries^{22,51-55}. nGDs were computed as the ratio of leaf-to-leaf branch length normalized by the total branch length of the phylogenetic trees generated by StrainPhlAn. These trees were constructed based on the marker gene alignments, specifically targeting positions with a minimum variability of 1%. For the SGB-specific nGD that were not estimated previously, the nGD value corresponding to the 3rd percentile of the distribution of nGD values among individuals was used. To enhance the reliability of strain identity threshold selection, we derived an alternative threshold for the 105 SGBs with $n_{\text{related}} \geq 5$ (i.e., detected in at least 5 sibling pairs). The species-specific strain identity threshold was calculated based on Youden's index and compared against the threshold obtained from the method described earlier; the more conservative of the two was selected as the final strain identity threshold. For the remaining strains with insufficient sharing events, we retained the SGB-specific nGD thresholds derived from previous studies or the 3rd percentile of the nGD distribution across individuals.

In detail, to evaluate the reproducibility of the species-specific strain-identity thresholds within our dataset, we included all 429 samples from related sibling pairs and unrelated subjects. This validation assessed the separation between the distributions of nGD distances for strains from the same SGB in these two groups. The analysis was performed for the 105 SGBs detected in at least 5 sibling pairs in this cohort. Given the absence of longitudinal samples, we defined true positives as pairwise phylogenetic distances (nGD) between sibling pairs that fell below the species-specific strain identity threshold—obtained either (1) from independent

longitudinal datasets of previous studies or as the 3rd percentile of the inter-individual nGD distribution, or (2) computed using Youden's index from our dataset. False positives were defined as nGD values from balanced unrelated pairs below the threshold, true negatives as those from balanced unrelated pairs above the threshold, and false negatives as nGD values from sibling pairs above the threshold. Performance metrics for the two thresholds in distinguishing strains from sibling pairs versus unrelated pairs were as follows: median precision = 0.97 and 0.98, and median F-score = 0.72 and 0.77 (Supplementary Figure 3; Supplementary Table S19).

Supplementary Figure 3

Response: Regarding the potential concern of co-ingested probiotic strains, the exclusion criteria of our study proactively addressed this confounder rather than relying on post hoc bioinformatic filtering. Furthermore, the top-ranked shared bacterial strains identified in our analysis are not typical food-grade or commercially used probiotic species, suggesting they are more likely to originate from environmental or communal household exposures.

Line 318: Exclusion criteria included the use of probiotics, antidepressants, anti-epileptics, or antibiotics within the one month prior to enrollment.

Reviewer #1:

Remarks to the Author:

Discussion: the conclusions about "shared pathogenic or beneficial strains" across siblings lack adequate technical rigor and may conflate co-acquisition with actual transmission. The statement that "cohabitation with ASD siblings worsens gut dysbiosis" implies directionality and causation in a study that is entirely cross-sectional. The possibility of reverse causation, shared environmental exposures, or common genetic susceptibility is acknowledged only briefly in limitations and not reflected in the interpretations drawn in results and discussion.

Response: We thank the reviewer for highlighting the importance of technical rigor and caution when interpreting strain-sharing data. In the revised manuscript, we have revised the manuscript to remove any language that might imply causation or directionality, ensuring our discussion remains strictly observational.

Specifically, we first note that children in ASD-M families show the most pronounced species-level dysbiosis and the highest strain-sharing rates with their siblings. We then report which bacterial strains are shared more frequently in ASD families compared to TD families. When discussing these strains, we focus on their annotated functions as potential traits of interest. We have also expanded the Discussion and Limitations sections to more thoroughly discuss other possible explanations. Such as shared environmental exposure, common genetic susceptibility, and have removed statements that could be misread as suggesting that cohabitation worsens dysbiosis. These revisions ensure our conclusions align with the cross-sectional nature of our study and improve the overall rigor of the manuscript. We are grateful for the reviewer's guidance in clarifying these points.

Line 193 (results): To evaluate gut microbiome sharing between siblings across different family types, we employed the chi-squared test with Yates' continuity correction or Fisher's exact test. Notably, several bacterial strains showed significantly higher sharing rates in multiplex ASD families compared to TD controls, including the opportunistic pathogens *Eubacterium rectale* (SGB4933), *Dorea formicigenerans* (SGB4575), and *Acidaminococcus intestini* (SGB5736) ($p_{\text{adj}} < 0.05$, Figure 4; Supplementary Table S20). In contrast, the sharing of *Bacteroides xylanisolvens* (SGB1867) was markedly lower in multiplex ASD families than in the other two groups. Interestingly, we also observed an increased prevalence of sharing for several common gut commensals in ASD families. For instance, *Bifidobacterium pseudocatenulatum* (SGB17237) and *Faecalibacterium prausnitzii* (SGB15342) exhibited their highest sharing percentages in multiplex ASD families, while *Sellimonas intestinalis* (SGB4617) showed a significantly increased sharing rate specifically in simplex ASD families ($p_{\text{adj}} < 0.05$, Figure 4). These differential sharing patterns underscore the significant influence of the familial context on the ASD microbiome. Rather than direct transmission alone, these findings suggest that the gut environment in children from ASD families—potentially shaped by shared environmental exposures and common genetic susceptibility—may be more receptive to the stable colonization of specific bacterial strains. This identifies the shared familial microbiome as a promising, adjustable target for future intervention strategies.

Line 256 (discussion): These observations suggest that the high strain-sharing rate observed in ASD-M families may be associated with cohabitation or their shared environmental exposures, particularly given that our initial species-level findings demonstrated more severe gut dysbiosis

and a higher abundance of opportunistic pathogens in ASD-M individuals. Furthermore, the differential bacteria identified between ASD-M and TD groups in our results showed correlative trends with ASD-related clinical scores, highlighting a potential clinical relevance. While we interpret the apparent reduced resistance in their gut microbiota as an observational finding, it may also be linked to potential common genetic susceptibility factors. Moreover, the sharing rates of three beneficial bacterial strains—*Faecalibacterium prausnitzii* (SGB15342), *Bifidobacterium pseudocatenulatum* (SGB17237), and *Sellimonas intestinalis* (SGB4617)—were significantly higher in ASD families compared to TD families. This indicates that intestinal susceptibility may not solely present disadvantages. The characteristic ease of colonization and high environmental transmissibility of these specific strains highlight a potential avenue for intervention strategies from a different perspective. These results point out the potential of microbiome-based therapeutics in alleviating the symptoms of ASD. Targeting shared beneficial bacteria in ASD families might be an effective intervention approach because these bacteria might be easily absorbed or colonized in the intestines.

Figure 4

Reviewer #1:

Remarks to the Author:

StrainPhlAn parameters and questionable settings:

Upon examining figure 5, an abnormal level of strain transmission is reported for two species which are normally found in the gut at a high abundance. Also, the Methods report that StrainPhlAn parameters were tuned to “marker_in_n_samples 1 -sample_with_n_markers 10”. In fact, StrainPhlAn does not have a parameter “marker_in_n_samples”, but has “marker_in_n_samples_perc”, which is normally set to 50, meaning only markers present in 50% of the samples are used. Tuning this parameter down to 1% would practically inflate the total diversity in the tree, potentially producing erroneous conclusions on strain similarity.

This parameter change increases the number of low-frequency markers in alignments, making downstream phylogenetic calls potentially unstable and distances less reliable.

Response: Thank you for the expert comment regarding the StrainPhlAn parameters. The reviewer correctly identified that we had used an inappropriately low value (marker_in_n_samples_perc 1) in our initial analysis. In accordance with this feedback, we have revised our methodology and re-ran the entire StrainPhlAn4 analysis pipeline using the recommended standard parameter (marker_in_n_samples_perc 50). This re-analysis,

conducted with the correct settings, confirmed that our core finding—significantly higher strain-sharing rates within ASD sibling pairs compared to TD sibling pairs—remains robust (Figure 3, please see below). While the adjustment resulted in subtle quantitative variations in the resulting phylogenetic trees and distances, leading to expected variations in certain intermediate results, such as specific p-values or strain-sharing rates for individual species, the primary conclusions of the study remain unchanged. We have updated the Methods section to accurately reflect the use of the correct parameter and value. We sincerely appreciate the reviewer’s vigilance, which has strengthened the methodological rigor of our strain-level analysis:

Line 388: Then, the strain-level profiling was performed with StrainPhlAn4 using the custom species-level genome bins (SGBs) marker database with the following parameters: "`marker_in_n_samples_perc 50` -sample_with_n_markers 10 -phylophlan_mode accurate -mutation_rates". The SGBs detected with "-print_clade_only" were selected for strain-level profiling.

Figure 3

Reviewer #1:

Remarks to the Author:

Also, while it is in principle “correct” to adopt previously established thresholds, these must be placed in the context of the same StrainPhlAn parameters used in the original publications. If the authors want to modify the algorithm settings, they must provide evidence that the updated parameters correctly segregate same-individual samples from inter-individual distances.

Response: We agree that strain-sharing thresholds should be validated under the specific StrainPhlAn parameters used. After correcting the parameter to the standard `marker_in_n_samples_perc 50` as noted in our previous response, we re-evaluated the performance of the original published thresholds alongside our cohort-derived thresholds calculated from the 3rd percentile of the distribution of nGD values. We also calculated new Youden-optimized thresholds where sufficient data were available. The performance of all

thresholds remained high, and the density plots confirmed clear separation between shared and non-shared distances. For downstream analysis, we consistently applied the most conservative threshold for each SGB. This re-validation under the corrected parameters ensures the methodological consistency and reliability of our strain-sharing conclusions. A more detailed description of this process, including the associated density plots (Supplementary Figure 3, please see the figure in the above response), has been provided in our response to your earlier comment on strain sharing assessment.

Reviewer #1:

Remarks to the Author:

Minor Concerns

The discussion on host genetics and microbiome convergence opens interesting avenues but remains speculative. No actual genetic data is presented, and the manuscript mixes correlation with mechanism throughout.

Dietary factors are briefly addressed via PCA, but only at a population level (no macronutrient or intake detail is shown), and these can be powerful confounders.

Technical details on sample storage, sequencing depth, and quality control metrics (e.g., coverage thresholds for strain calling) are minimal; replications or reproducibility checks are not discussed.

Response: We thank the reviewer for recognizing the value of our study and for noting the limitation regarding the lack of genetic data. In our subsequent follow-up research, we have collected relevant biological samples and will consider conducting genetic-related studies. Regarding dietary factors, in addition to the overall PCA analysis, we have incorporated further adjustments as detailed below:

Line 125: Furthermore, although the overall dietary patterns showed no significant differences between groups based on PCA analysis of daily diet intake (Supplementary Figure 1), we have conducted sensitivity analyses adjusting for dietary factors. First, we performed Kruskal-Wallis tests for overall differences of dietary factors (Supplementary Table S2), followed by pairwise Wilcoxon tests with FDR correction for each significantly different factor (Supplementary Table S3). After incorporating these significant dietary factors into Maaslin2 models, the main findings remained robust. Specifically, 10 and 3 differentially abundant bacterial species were associated with ASD-M and ASD-S, respectively (Supplementary Figure 2b and 2h), and 2 associated species emerged in the ASD-O group (Supplementary Figure 2n).

Line 605: (Figure legend) **Supplementary Figure 1. Principal component analysis (PCA) of dietary daily intake among individuals from different groups.** Dietary patterns assessed by daily food consumption. Kruskal–Wallis test, PC1, $p=0.198$. PC2, $p=0.0867$.

Line 609: (Figure legend) **Supplementary Figure 2. Differential bacterial species associated with ASD in different family types.** Associations between gut microbiota and children from different family types (ASD-M, ASD-S, ASD-O, TD) were assessed using multivariate linear models (MaAsLin2; significance: $p < 0.05$, FDR < 0.2). (a), (c), (e), (g), (i), (k), (m): Comparisons of ASD family types versus TD, and stratified analyses within the same family type by ADHD status (+/-), adjusting for confounders except dietary factors; (b), (d), (f), (h), (j), (l), (n): The same comparisons, adjusting for confounders including dietary factors.

Supplementary Figure 1

Diet PCA (PC1 $p=0.198$, PC2 $p=0.0867$)

Supplementary Figure 2

Reviewer #1:

Remarks to the Author:

The figures are not as detailed as the complexity of the study design warrants. For example, no visualization of strain-sharing networks or transmission matrices is provided. The authors should specify which MetaPhlAn/StrainPhlAn database was used (and which version, for transparency and reproducibility).

Response: We sincerely appreciate the reviewer's suggestion to enhance the visualization of the study's complexity, particularly concerning strain-sharing dynamics. To address this, we have significantly expanded our supplementary figures to provide the requested detailed visualization and analysis:

Visualization of Strain-Sharing: We have added a comprehensive visualization of the strain-sharing network (Supplementary Figure 4b), which depicts the connections among individuals based on shared strains.

To further support our findings on the distinct patterns across family types, we have also added two complementary visualizations: A strain-sharing Density Plot (Supplementary Figure 4a) to show the distribution of sharing rates and a Clustering Coefficient Plot (Supplementary Figure 4c) derived from the network topology analysis, which quantitatively demonstrates the highly interconnected structure within Multiplex ASD families.

These new figures and the associated network topology metrics confirm that strain transmission in Multiplex ASD families is more robust and pervasive, aligning with the pattern of dysbiosis observed.

Line 177: Consistent with previous observations that microbiome transmission occurs beyond close contacts and reflects population structure, we observed distinct strain-sharing patterns across family types. This was evidenced by both the density distributions of sharing rates (Supplementary Figure 4a, Supplementary Table S17) and a PERMANOVA performed on the strain-sharing network (Euclidean distance matrix; $n = 747$, $R^2 = 10.6\%$, $p = 0.001$; Supplementary Figure 4b and Methods). Network topology analysis revealed that the extensive distribution of Multiplex families in the visualization corresponds to a highly interconnected structure rather than a loose association, followed by simplex ASD families, and then TD families. Sibling pairs from multiplex ASD families exhibited significantly higher clustering coefficient compared to other families (Kruskal-Wallis test, $p = 0.00053$, $\chi^2 = 15.098$; Supplementary Figure 4c), suggesting the formation of tight-knit strain-sharing communities. These topological metrics confirm that strain transmission in multiplex ASD families is more robust and pervasive.

Line 432: (Methods) **Strain-sharing networks**

Unsupervised strain-sharing networks were constructed and visualized using the R packages tidygraph (v1.2.3), igraph (v1.4.3), and plotted with ggraph (v2.1.0). Networks were laid out using a stress-minimization algorithm. Nodes represent individuals, and edges connect pairs that share ≥ 10 bacterial strains. Only edges meeting this threshold are displayed. We calculated the Local Clustering Coefficient for each node using the igraph package in R.

a

b

c

Response: Thank you for this suggestion regarding the database and version. In response, we have updated the Methods section to specify the exact database versions used for reproducibility.

Line 382: **Microbial taxonomic profiles**

Microbial composition profiles were analyzed from the quality-filtered forward reads using MetaPhlAn4⁴⁹ (version 4.1.4) with default parameters and the vOct22_202212 database. To accelerate processing, all analyses were run in parallel using GNU parallel (version 2018). Taxonomic profiling at the strain levels was performed using StrainPhlAn4⁵⁰ (version 4.1.4) with default parameters.

Then, the strain-level profiling was performed with StrainPhlAn4 using the custom species-level genome bins (SGBs) marker database with the following parameters: "marker_in_n_samples_perc 50 -sample_with_n_markers 10 -phylophlan_mode accurate -mutation_rates". The SGBs detected with "-print_clade_only" were selected for strain-level profiling. This marker-gene-based approach has been benchmarked to provide robust strain-level identification even at an average coverage of 1.3× per SGB, consistently outperforming assembly-based methods in samples with moderate sequencing depth.

Reviewer #2:

Remarks to the Author:

Weaknesses include: 1) The different families showed significant differences in age, sex and ADHD diagnosis (unclear why ADHD was assessed). Likely differences in factors important in ASD symptomatology and clinical presentation including dietary habits, number of pets, ASD severity, GI symptoms were not addressed.

Response: We sincerely appreciate the reviewer's insightful comments regarding host factors and potential confounders. We have addressed these concerns point-by-point below:

1.Regarding Age, Sex, and ADHD: ADHD was assessed and included in our analysis due to its high comorbidity rate with ASD. To rigorously account for the demographic differences observed, we treated age, sex, and ADHD diagnosis as covariates in our statistical models. Furthermore, we conducted stratified analyses based on ADHD status to ensure our findings were not driven by this comorbidity. The results remained robust across these adjustments.

2.Regarding Dietary Habits: We concur that diet is a critical factor. We performed a PCA analysis of overall dietary patterns, which showed no significant clustering or differences between the groups (Supplementary Figure 1). Additionally, we performed sensitivity analyses adjusting for dietary factors, which further confirmed the stability of our results (Supplementary Figure 2).

Line 125: Furthermore, although the overall dietary patterns showed no significant differences between groups based on PCA analysis of daily diet intake (Supplementary Figure 1), we have conducted sensitivity analyses adjusting for dietary factors. First, we performed Kruskal-Wallis tests for overall differences of dietary factors (Supplementary Table S2), followed by pairwise Wilcoxon tests with FDR correction for each significantly different factor (Supplementary Table S3). After incorporating these significant dietary factors into Maaslin2 models, the main findings remained robust. Specifically, 10 and 3 differentially abundant bacterial species were associated with ASD-M and ASD-S, respectively (Supplementary Figure 2b and 2h), and 2 associated species emerged in the ASD-O group (Supplementary Figure 2n).

Supplementary Figure 2:

Supplementary Figure 2

3.Regarding Other Environmental Factors (e.g., Pets): We acknowledge the reviewer’s valid point regarding other potential confounders, such as the number of pets, which were not included in our original questionnaire. While we comprehensively adjusted for measured confounders (age, sex, ADHD, and diet), we recognize that data on certain environmental exposures were unavailable. We have explicitly addressed this constraint in the revised Limitation section of the manuscript to ensure transparency regarding potential unmeasured confounding.

Line 300: Finally, while we meticulously adjusted for collected covariates, observational human microbiome studies are inherently subject to unmeasured confounders. We lacked data on specific environmental factors, such as the number of pets in the household.

**4. Concerning ASD severity**, given the various complex measurements and standards for ASD-related clinical scores, we were concerned that including all scores in the analytical models might lead to overfitting. Therefore, we conducted additional correlation analyses between differential bacteria identified in the ASD-M vs. TD comparison and ASD-related scores.

Line 215: **Gut microbiome is associated with ASD clinical scores**

To extend the findings and explore the clinical relevance. We processed the association analysis of the markers and clinical parameters. We found 7 multiplex ASD families enriched markers based on the MaAslin2 were significantly positively correlated with several ASD-related clinical parameters, including Social Responsiveness Scale (SRS), Social Experience Questionnaire (SEQ), Child Behaviour Checklist (CBCL), Anxiety scale for children-ASD (ASC-ASD) (Spearman's correlation, $p<0.05$, Supplementary Figure 5), including *Coprobacillus cateniformis*, *Murimonas intestini*, *Eisenbergiella tayi*, *Alistipes finegoldii*, *Lachnospira pectinoschiza*, *Sanguibacter SGB15121*, and *Eubacterium siraeum*. In contrast, the TD enriched species *Faecalibacterium prausnitzii*, GGB51510_SGB71883, *Agathobaculum butyriciproducens*, and *Bacteroides xylanisolvens* were negatively correlated with the ASD clinical parameters (Spearman's correlation, $p<0.05$, Supplementary Figure 5)

Line 238: Our correlation analysis also indicated that differential bacteria between ASD-M and TD are associated with ASD-related scores, thereby enhancing the reference value of our clinical observations.

Supplement Figure 5:

5. Regarding the GI symptoms: In our cohort, the proportion of children with GI symptoms was below 15%, so we considered its potential impact to be limited and did not perform specific adjustments.

Reviewer #2:

Remarks to the Author:

2. Results do not prove a causal role of altered gut microbial composition in ASD, and other than confirming the lateral transmission of gut microbial features.

In summary, despite the size of the study and high quality of analyses, I do not see a major progress or an actionable finding from this well executed study for the treatment of children with ASD.

Response: Thank you for your valuable feedback regarding the clinical and mechanistic implications of our work. We sincerely appreciate the positive assessment of our study's quality and size. We have refined our discussion to address the points. We fully acknowledge that, as a cross-sectional observational study, our data cannot establish a causal link between the altered gut microbiome and ASD. Our primary aim was to characterize the distinct gut microbial patterns and differing degrees of bacterial strain sharing across diverse ASD family types. The gut microbiome shows the most significant dysbiosis at the species level in ASD children from Multiplex families. Strain sharing is highest in Multiplex ASD sibling pairs, intermediate in Simplex pairs, and lowest in TD pairs. This pattern underscores the influence of the familial context on the ASD microbiome. While this does not prove causation, it strongly suggests that the gut environment in some children with ASD may be more receptive to colonization by certain bacterial strains within a shared household environment. While we acknowledge that the distance between identifying these microbial features and developing actionable treatments is considerable, our findings nonetheless establish the gut microbiota as a promising, adjustable target. This observation could inform future studies exploring whether modifying the shared microbial environment within families represents a viable intervention strategy. We believe these adjustments provide a clearer and more measured interpretation of our findings, and we appreciate your guidance in strengthening the manuscript.

Reviewer #3:

Remarks to the Author:

Response: We thank the reviewer for their participation in the peer-review process. We sincerely appreciate the time and effort invested in evaluating our manuscript, and we are grateful for the valuable contribution made through this collaborative review initiative by Nature Communications.

POINT-BY-POINT REPLY TO EDITORS AND REVIEWERS

Dear reviewers,

We are deeply grateful for your continued engagement and for the additional insightful feedback, which has further strengthened our work, particularly in refining the precision of our terminology. We have provided a point-by-point response to all comments below. The revised manuscript is attached in both tracked and clean versions, with all changes highlighted in yellow for your convenience.

We appreciate the opportunity to resubmit and hope this revision is acceptable for publication.

Yours sincerely

Siew Ng on behalf of co-authors

Reviewers' Comments:

Reviewer #1:

Remarks to the Author:

1. Causal and directional language

Rephrase any statements implying influence, susceptibility, receptivity, or directionality to strictly associational language, as no causal inference can be made from the current study design.

Section: Results (strain-sharing summary)

Sentence to be replaced or modified:

“These differential sharing patterns underscore the significant influence of the familial context on the ASD microbiome.”

Suggested replacement:

“These differential sharing patterns underscore a strong association between familial context and gut microbiome configuration in ASD.”

Response: We sincerely appreciate the reviewer's critical reminder regarding the causal and directional language. We fully agree that as a cross-sectional study, our data support associations rather than causal influences. We have replaced the word "influence" with "strong association" and "ASD microbiome" with "gut microbiome configuration in ASD," exactly as suggested.

Line 211: **These differential sharing patterns underscore a strong association between familial context and gut microbiome configuration in ASD.**

2. Strain-sharing interpretation (transmission vs association)

Strain sharing observed in a cross-sectional cohort cannot distinguish direct microbial transmission from co-acquisition or convergence and should therefore be framed as association or co-occurrence only.

Section: Results (strain-level comparison across family types)

Sentence to be replaced:

“These findings indicate that the gut environment in children from ASD families facilitates the transmission of specific bacterial strains.”

Suggested replacement or modified:

“These findings indicate that children from ASD families exhibit distinct strain-sharing patterns compared with TD families, without implying direct microbial transmission.”

4. Host “susceptibility” or “receptivity” framing

Phrasing that implies intrinsic host susceptibility or biological receptivity is not supported without direct host-level functional measurements.

Section: Discussion (interpretation of strain-sharing patterns)

Sentence to be replaced or modified:

“The gut environment in children from ASD families may be more receptive to the stable colonization of specific bacterial strains.”

Suggested replacement:

“The gut microbiome configurations observed in children from ASD families may be associated with increased persistence or detectability of specific bacterial strains.”

Response: We sincerely thank the reviewer for these critical insights. We agree that a cross-sectional study design limits our ability to distinguish between direct transmission and other ecological processes. As suggested, we have revised the text in the Results section. This new phrasing accurately reflects the associative nature of our data while avoiding unsupported implications of direct microbial transmission or intrinsic host receptivity.

In accordance with your suggestions from comment 2 & 4, we have:

1. Re-framed our findings in the Results as "distinct strain-sharing patterns" rather than "facilitating transmission."
2. Replaced terms like "receptivity" with more objective descriptions such as "increased persistence or detectability" of specific strains.
3. Clarified that these configurations are "associated with" microbiome patterns without implying a direct change in microbial transmission.

Line 212: Rather than direct transmission, these findings indicate that children from ASD families exhibit distinct strain-sharing patterns compared with TD families. These patterns, which may be influenced by shared environmental and host factors, suggest that the gut microbiome configurations in ASD families are associated with the increased persistence or detectability of specific bacterial strains.

3. Use of “opportunistic pathogen” terminology

Labeling taxa as pathogens risks clinical overinterpretation, particularly when pathogenicity is context-dependent and not assessed in this study.

Section: Results (description of shared taxa)

Sentence to be replaced or modified:

“including the opportunistic pathogens *Eubacterium rectale*, *Dorea formicigenerans*, and *Acidaminococcus intestini*.”

Suggested replacement:

“including taxa with reported opportunistic or context-dependent pathogenic potential, such as *Eubacterium rectale*, *Dorea formicigenerans*, and *Acidaminococcus intestini*.”

Response: We agree with the reviewer that the pathogenicity of these taxa is context-dependent and was not directly assessed in this study. To avoid clinical overinterpretation, we have adopted the more nuanced phrasing suggested by the reviewer to describe these taxa.

Line 199: Notably, several bacterial strains showed significantly higher sharing rates in multiplex ASD families compared to TD controls, including taxa with reported opportunistic or context-dependent pathogenic potential, such as *Eubacterium rectale* (SGB4933), *Dorea formicigenerans* (SGB4575), and *Acidaminococcus intestini* (SGB5736) ($p_{\text{adj}} < 0.05$, Figure 4; Supplementary Table S20).

Line 259: This elevated sharing is characterized by a group of microorganisms typically recognized for their opportunistic or context-dependent pathogenic potential, including *Eubacterium rectale* (SGB4933), *Dorea formicigenerans* (SGB4575) and *Acidaminococcus intestini* (SGB5736).

5. Therapeutic and interventional implications

Translational or interventional relevance cannot be inferred from cross-sectional observational data and must be framed as hypothesis-generating only.

Section: Discussion (future directions or concluding paragraph)

Sentence to be replaced or modified:

“This identifies the shared familial microbiome as a promising, adjustable target for future intervention strategies.”

Suggested replacement:

“This identifies the shared familial microbiome as a hypothesis-generating context that may be relevant for future interventional studies, pending longitudinal and mechanistic validation.”

Response: We completely agree that translational implications must be approached with caution in the absence of longitudinal or mechanistic data. We have revised our conclusion to emphasize that our findings serve as a foundation for hypothesis generation rather than direct clinical recommendations.

Line 217: This identifies the shared familial microbiome as a hypothesis-generating context that may be relevant for future interventional studies, pending longitudinal and mechanistic validation.

6. Genetics discussion

Host genetic contributions are discussed without corresponding genetic data and must be clearly identified as speculative.

Section: Discussion (genetics paragraph)

Response: We agree that host genetic contributions remain speculative without direct sequencing data. We have revised the relevant paragraph in the Discussion to explicitly identify these points as hypotheses. Specifically, we have added a clarifying statement that "host genetics were not directly assessed in the current study" and ensured that all related conclusions are framed using speculative language (e.g., "might potentially" and "remain speculative").

Line 297: In the context of our study, host genetics were not directly assessed in the current progress. Therefore, we speculatively suggest that a shared genetic background in multiplex families might potentially contribute to the observed strain-sharing patterns and microbiome convergence, alongside shared environmental exposures. However, without corresponding host functional or genetic data, these potential links remain speculative. Further integrative studies are needed to elucidate the potential associations between host genetic backgrounds and the heterogeneity of the gut microbiome across different ASD family types.

7. Directionality and causation disclaimer

The manuscript should include an explicit and unambiguous statement clarifying that directionality cannot be inferred, as from a cross-sectional study, one cannot distinguish between direct microbial transmission, co-acquisition from shared environments, or microbiome convergence driven by unmeasured host factors, and no inference regarding directionality can be made.

Section: Limitations

Also: As all participants were Hong Kong Chinese, the generalisability of these findings to other populations remains to be established

Response: We have added an explicit and unambiguous disclaimer in the Limitations section to address the inherent constraints of our study design. The revised text now clarifies that our study cannot distinguish between "direct microbial transmission, co-acquisition from shared environments, or microbiome convergence driven by unmeasured host factors." Furthermore, we have highlighted the limited generalizability of our findings, noting that the cohort consisted exclusively of Hong Kong Chinese participants.

Line 309: Crucially, no inference regarding the directionality of our findings can be made. As a cross-sectional study, our design cannot distinguish between direct microbial transmission, co-acquisition from shared environments, or microbiome convergence driven by unmeasured host factors or common genetic susceptibility; therefore, directionality and causality cannot be inferred. Second, while we adjusted for collected covariates, observational human microbiome studies are inherently subject to unmeasured confounders, such as specific household environmental factors (e.g., the number of pets) that were not recorded. Finally, as all participants in this study were Hong Kong Chinese, the generalizability of these findings to other ethnic or geographical populations remains to be established. Further longitudinal studies and mechanistic investigations are required to distinguish between these ecological processes and validate our findings in more diverse cohorts.

Reviewer #3:

Remarks to the Author:

Response: We thank the reviewer for their participation in the peer-review process as part of the Nature Communications co-review initiative. We sincerely appreciate the professional and constructive feedback provided by the team.